# A novel lineage of candidate pheromone receptors for sex communication in moths

**Lucie Bastin-Héline[1], Arthur de Fouchier[1†], Song Cao[2], Fotini Koutroumpa[1], Gabriela Caballero-Vidal[1], Stefania Robakiewicz[1], Christelle Monsempes[1], Marie-Christine François[1], Tatiana Ribeyre[1], Annick Maria[1], Thomas Chertemps[1], Anne de Cian[3], William B Walker III[4], Guirong Wang[2]\*, Emmanuelle Jacquin-Joly[1]\*, Nicolas Montagné[1]\***

[1]Sorbonne Université, Inra, CNRS, IRD, UPEC, Université Paris Diderot, Institute of Ecology and Environmental Sciences of Paris, Paris and Versailles, France; [2]State Key Laboratory for Biology of Plant Diseases and Insect Pests, Institute of Plant Protection, Chinese Academy of Agricultural Sciences, Beijing, China; [3]CNRS UMR 7196, INSERM U1154, Museum National d'Histoire Naturelle, Paris, France; [4]Department of Plant Protection Biology, Swedish University of Agricultural Sciences, Alnarp, Sweden

**Abstract** Sex pheromone receptors (PRs) are key players in chemical communication between mating partners in insects. In the highly diversified insect order Lepidoptera, male PRs tuned to female-emitted type I pheromones (which make up the vast majority of pheromones identified) form a dedicated subfamily of odorant receptors (ORs). Here, using a combination of heterologous expression and in vivo genome editing methods, we bring functional evidence that at least one moth PR does not belong to this subfamily but to a distantly related OR lineage. This PR, identified in the cotton leafworm *Spodoptera littoralis*, is highly expressed in male antennae and is specifically tuned to the major sex pheromone component emitted by females. Together with a comprehensive phylogenetic analysis of moth ORs, our functional data suggest two independent apparitions of PRs tuned to type I pheromones in Lepidoptera, opening up a new path for studying the evolution of moth pheromone communication.

**\*For correspondence:**
wangguirong@caas.cn (GW);
emmanuelle.joly@inra.fr (EJ-J);
nicolas.montagne@sorbonne-
universite.fr (NM)

**Present address:** †Laboratoire d'Ethologie Expérimentale et Comparée (LEEC), Université Paris 13, Sorbonne Paris Cité, Villetaneuse, France

## Introduction

The use of pheromone signals for mate recognition is widespread in animals, and changes in sex pheromone communication are expected to play a major role in the rise of reproductive barriers and the emergence of new species (*Smadja and Butlin, 2009*). Since the first chemical identification of such a pheromone in *Bombyx mori* (*Butenandt et al., 1959*), moths (Insecta, Lepidoptera) have been a preferred taxon for pheromone research (*Cardé and Haynes, 2004*; *Kaissling, 2014*). The diversification of pheromone signals has likely played a prominent role in the extensive radiation observed in Lepidoptera, which represents almost 10% of the total described species of living organisms (*Stork, 2018*).

Female moths generally release a species-specific blend of volatile molecules, which attract males over a long distance (*Cardé and Haynes, 2004*). Four types of sex pheromones have been described in moths (types 0, I, II and III), with distinct chemical structures and biosynthetic pathways (*Löfstedt et al., 2016*). 75% of all known moth sex pheromone compounds belong to type I and are straight-chain acetates, alcohols or aldehydes with 10 to 18 carbon atoms (*Ando et al., 2004*). Type I pheromones have been found in most moth families investigated, whereas the other types are restricted to only a few families (*Löfstedt et al., 2016*).

**eLife digest** Many animals make use of chemical signals to communicate with other members of their species. Such chemical signals, called pheromones, often allow males and females of the same species to recognize each other before mating. Since the discovery of the very first pheromone in the silkworm moth *Bombyx mori* at the end of the 1950s, moths have been a model for pheromone research. The sex pheromone communication system in these insects has thus been well described: females emit a mixture of volatile chemicals, which can be detected by the antennae of males up to several hundred meters away. This detection is achieved through neurons with specialized proteins known as pheromone receptors that bind to the chemical signals produced by the females.

Recognizing mates by detecting a very specific pheromone signature prevents moths from interbreeding with other species. The evolution of pheromone signals and their corresponding receptors can therefore lead to the rise of new reproductive barriers between populations, and eventually to the emergence of new species. The rate at which sex pheromones have diversified is likely one reason for the existence of over 160,000 species of moths. But how did moths' sex pheromone receptors evolve in the first place?

Previous studies suggested that moth pheromone receptors had appeared just once during evolution. Specifically, they revealed that these receptors belong to the same branch or lineage in the 'family tree' of all receptors that detect chemical compounds in moths. This meant that when researchers looked for pheromone receptors in a new species of moth, they always focused on this lineage. But Bastin-Héline et al. have now found that one pheromone receptor from a pest moth called *Spodoptera littoralis* does not belong to this established group.

First, Bastin-Héline et al. inserted this receptor into animal cells grown in the laboratory to confirm that it responds to a specific pheromone produced by *S. littoralis.* Next, they genetically modified moths of this species and showed that males need this receptor in order to mate. An evolutionary analysis showed that the receptor belongs to a different lineage than all the other known pheromone receptors. Together these results indicate the receptors for sex pheromones must have evolved multiple times independently in moths.

These results will open new avenues for deciphering pheromone communication in moths, and lead to further research into this newly discovered lineage of candidate pheromone receptors. Such studies may foster the development of new strategies to control agricultural pests, given that some species of moths can have devastating effects on the yields of certain crops.

In moth male antennae, pheromone compounds are detected by dedicated populations of olfactory sensory neurons (OSNs). Each type of OSN usually expresses one pheromone receptor (PR) responsible for signal transduction. PRs are 7-transmembrane domain proteins belonging to the odorant receptor (OR) family and, as ORs, are co-expressed in OSNs together with the conserved co-receptor Orco (*Chertemps, 2017*; *Fleischer and Krieger, 2018*).

Since the first discovery of moth PRs (*Krieger et al., 2004*; *Sakurai et al., 2004*), numerous pheromone receptors tuned to type I pheromone compounds have been characterized through different hererologous expression systems, and most appeared to be specific to only one compound (*Zhang and Löfstedt, 2015*). More recently, a few receptors tuned to type 0 and type II pheromones have also been characterized (*Zhang et al., 2016*; *Yuvaraj et al., 2017*). Type I PRs belong to a dedicated monophyletic subfamily of ORs, the so-called 'PR clade', suggesting a unique emergence early in the evolution of Lepidoptera (*Yuvaraj et al., 2018*). Another hallmark of type I PRs is their male-biased expression (*Koenig et al., 2015*). The phylogenetic position and the expression pattern have thus been the main criteria used to select candidate PRs for functional studies.

Likewise, we selected PRs from the male transcriptome of the cotton leafworm *Spodoptera littoralis* (*Legeai et al., 2011*), a polyphagous crop pest that uses type I sex pheromone compounds (*Muñoz et al., 2008*) and that has been established as a model in insect chemical ecology (*Ljungberg et al., 1993*; *Binyameen et al., 2012*; *Saveer et al., 2012*; *Poivet et al., 2012*; *de Fouchier et al., 2017*). Through heterologous expression, we characterized two PRs tuned to minor components of the *S. littoralis* pheromone blend (*Montagné et al., 2012*; *de Fouchier et al., 2015*), but none of the tested candidate PRs detected the major pheromone component (*Z,E*)-9,11-

tetradecadienyl acetate (hereafter referred as (Z,E)-9,11-14:OAc), which is necessary and sufficient to elicit all steps of the male mate-seeking behavioral sequence (*Quero et al., 1996*).

In order to identify new type I PR candidates, we focused on male-biased ORs, whether they belong to the PR clade or not. Notably, a preliminary analysis of *S. littoralis* OR expression patterns led to the identification of such a receptor, SlitOR5, which was highly expressed in male antennae but did not belong to the PR clade (*Legeai et al., 2011*). Furthermore, a recent RNAseq analysis showed that SlitOR5 was the most abundant OR in *S. littoralis* male antennae (*Walker et al., 2019*). Here, we demonstrate that SlitOR5 is the receptor for (Z,E)-9,11-14:OAc using a combination of heterologous expression and in vivo genome editing methods. Based on a comprehensive phylogenetic analysis of lepidopteran ORs, we show that SlitOR5 belongs to an OR subfamily that is distantly related to the PR clade but harbors numerous sex-biased ORs from distinct moth families. Altogether, these results suggest that PRs detecting type I pheromones evolved at least twice in Lepidoptera, which offers a more detailed and complex panorama on moth PR evolution.

## Results

### *SlitOr5* is highly expressed in males but does not belong to the type I pheromone receptor clade

We first used quantitative real-time PCR to compare the relative expression levels of the *SlitOr5* gene in *S. littoralis* male and female adult antennae. We found *SlitOr5* expressed with a more than

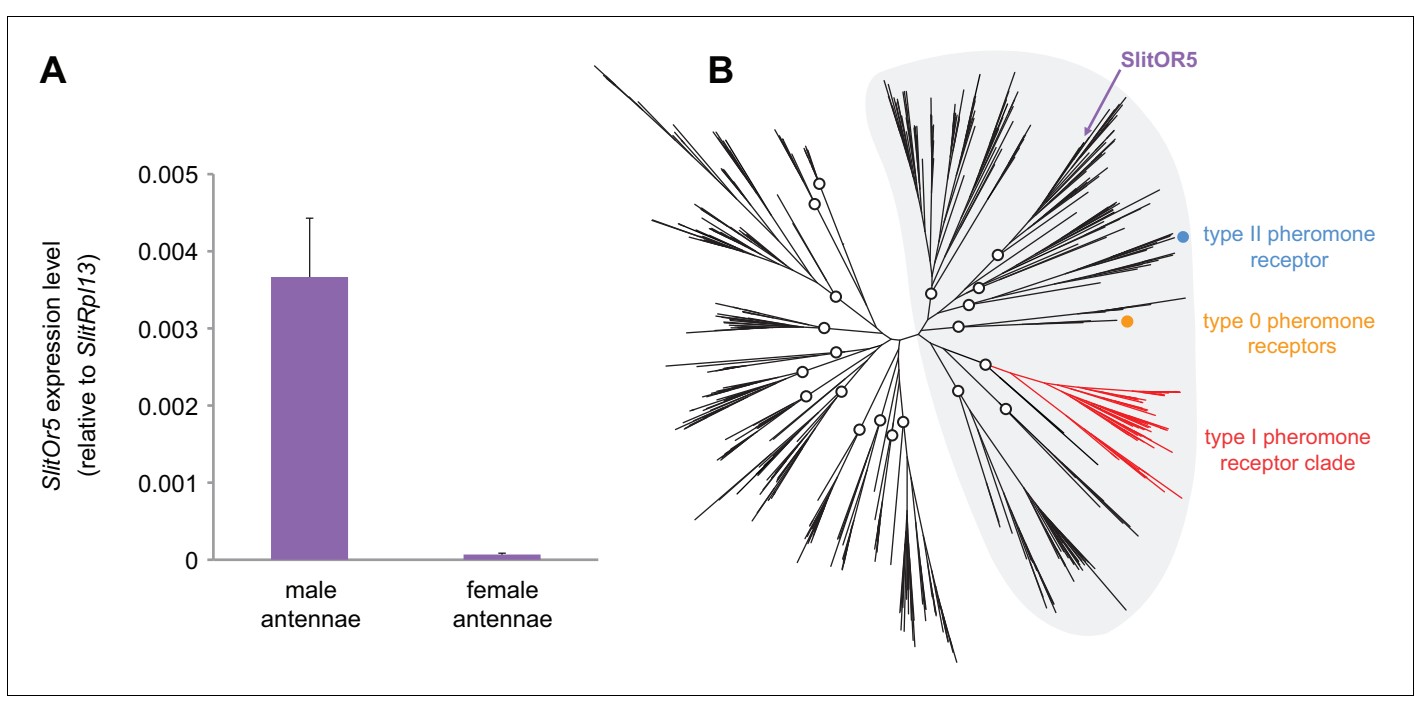

**Figure 1.** *SlitOr5* is highly expressed in males but does not belong to the pheromone receptor clade. (**A**) Expression levels of *SlitOr5* in adult male and female antennae of *S. littoralis*, as measured by real-time qPCR. Expression levels have been normalized to the expression of *SlitRpl13*. Plotted values represent the mean normalized expression values ± SEM (*n* = 3). Raw results are available in *Figure 1—source data 1*. (**B**) Unrooted maximum likelihood phylogeny of lepidopteran ORs, based on 506 amino acid sequences from nine species, each belonging to a different superfamily. The position of SlitOR5 and of receptors for type 0, type I and type II pheromone compounds is highlighted. Circles on the nodes indicate the distinct paralogous OR lineages, supported by a transfer bootstrap expectation (TBE) >0.9. All the PR-containing lineages grouped within a large clade (highlighted in grey) also supported by the bootstrap analysis. The sequence alignment file is available in *Figure 1—source data 2*.

The online version of this article includes the following source data for figure 1:

**Source data 1.** Mean normalized expression values of *SlitOr5* measured in the three biological replicates.
**Source data 2.** Alignment of amino acid sequences used to build the phylogeny (FASTA format).

50-fold enrichment in the male antennae (*Figure 1A*), thus confirming previous observations (*Legeai et al., 2011*; *Walker et al., 2019*).

We reconstructed a maximum likelihood phylogeny of lepidopteran ORs, based on entire OR repertoires from nine different species. Among the 20 paralogous lineages identified (each having evolved in principle from an ancestral OR present in the last common ancestor of Lepidoptera), SlitOR5 belonged to a lineage distantly related to the type I PR clade, as well as to the lineages containing type 0 and type II PRs (*Figure 1B*). These four paralogous lineages grouped within a larger clade highly supported by the bootstrap analysis (highlighted in grey in *Figure 1B*). This clade has been previously shown to contain ORs tuned to terpenes and aliphatic molecules – including sex pheromones – and exhibits higher evolutionary rates compared to more ancient clades that contain many receptors for aromatics (*de Fouchier et al., 2017*).

## SlitOR5 binds (*Z,E*)-9,11-14:OAc with high specificity and sensitivity

We next used two complementary heterologous systems to characterize the function of SlitOR5 and assess whether it is the receptor to (*Z,E*)-9,11-14:OAc, the major component of the *S. littoralis* sex pheromone blend. First, we expressed SlitOR5 in *Drosophila melanogaster* OSNs housed in at1 trichoid sensilla, in place of the endogenous PR DmelOR67d (*Kurtovic et al., 2007*). Single-sensillum recordings were performed to measure the response of at1 OSNs to 26 type I pheromone compounds (*Supplementary file 1*), including all the components identified in the pheromones of *Spodoptera* species (*El-Sayed, 2018*). SlitOR5-expressing OSNs strongly responded to (*Z,E*)-9,11-14:OAc ($65 \pm 15$ spikes.s$^{-1}$), whereas there was no significant response to any other compound (*Figure 2A*).

Then, we co-expressed SlitOR5 with its co-receptor SlitOrco in *Xenopus* oocytes and recorded the responses to the same panel of pheromone compounds using two-electrode voltage-clamp. A strong current was induced in SlitOR5-expressing oocytes when stimulated with (*Z,E*)-9,11-14:OAc ($3.9 \pm 0.3$ μA), whereas only minor currents were recorded in response to (*Z,E*)-9,12-14:OAc and (*Z*) 9-12:OAc (*Figure 2B*). SlitOR5 sensitivity was assessed with a dose-response experiment that showed a low detection threshold ($10^{-8}$ M) and an EC$_{50}$ of $1.707 \times 10^{-7}$ M (*Figure 2D–E*).

We compared the response spectra of heterologously expressed SlitOR5 with that of *S. littoralis* male OSNs housed in type one long trichoid sensilla (LT1A OSNs, *Figure 2C*), known to detect (*Z,E*)-9,11-14:OAc (*Ljungberg et al., 1993*; *Quero et al., 1996*). When stimulated with the 26 pheromone compounds, LT1A OSNs significantly responded to (*Z,E*)-9,11-14:OAc ($55 \pm 4$ spikes.s$^{-1}$) and to a lesser extent to its stereoisomer (*Z,Z*)-9,11-14:OAc, which is absent from any *Spodoptera* pheromone. This mirrored the response spectra of heterologously expressed SlitOR5, especially the one observed in *Drosophila* OSNs (*Figure 2A*).

## In vivo response to (*Z,E*)-9,11-14:OAc is abolished in *SlitOr5* mutant males

To confirm in vivo that SlitOR5 is the receptor for the major sex pheromone component of *S. littoralis*, we carried out a loss-of-function study by generating mutant insects for the gene *SlitOr5*. The CRISPR/Cas9 genome editing system was used to create a mutation in the first exon of *SlitOr5* with the aim of disrupting the open-reading frame. Guide RNAs were injected along with the Cas9 protein in more than one thousand eggs. We genotyped 66 G0 hatched larvae and found seven individuals carrying at least one mutation in *SlitOr5*. These 7 G0 individuals were back-crossed with wild-type individuals to create 7 G1 heterozygous mutant lines. We selected a line carrying a single mutation that consisted of a 10 bp deletion at the expected location, introducing a premature STOP codon within the *SlitOr5* transcript after 247 codons (*Figure 3A*).

We next generated G2 homozygous mutant males (*SlitOr5*$^{-/-}$) and compared their ability to detect (*Z,E*)-9,11-14:OAc to that of wild-type and heterozygous (*SlitOr5*$^{+/-}$) males, using electroantennogram (EAG) recordings (*Figure 3B*). When stimulated with (*Z,E*)-9,11-14:OAc, wild-type and *SlitOr5*$^{+/-}$ antennae exhibited similar EAG amplitudes (0.89 mV and 1.16 mV, respectively), whereas the response was completely abolished in *SlitOr5*$^{-/-}$ antennae (0.02 mV). Control experiments using a *S. littoralis* minor pheromone component and plant volatiles known to induce EAG responses in *S. littoralis* (*Saveer et al., 2012*; *López et al., 2017*) showed that antennal responses were not

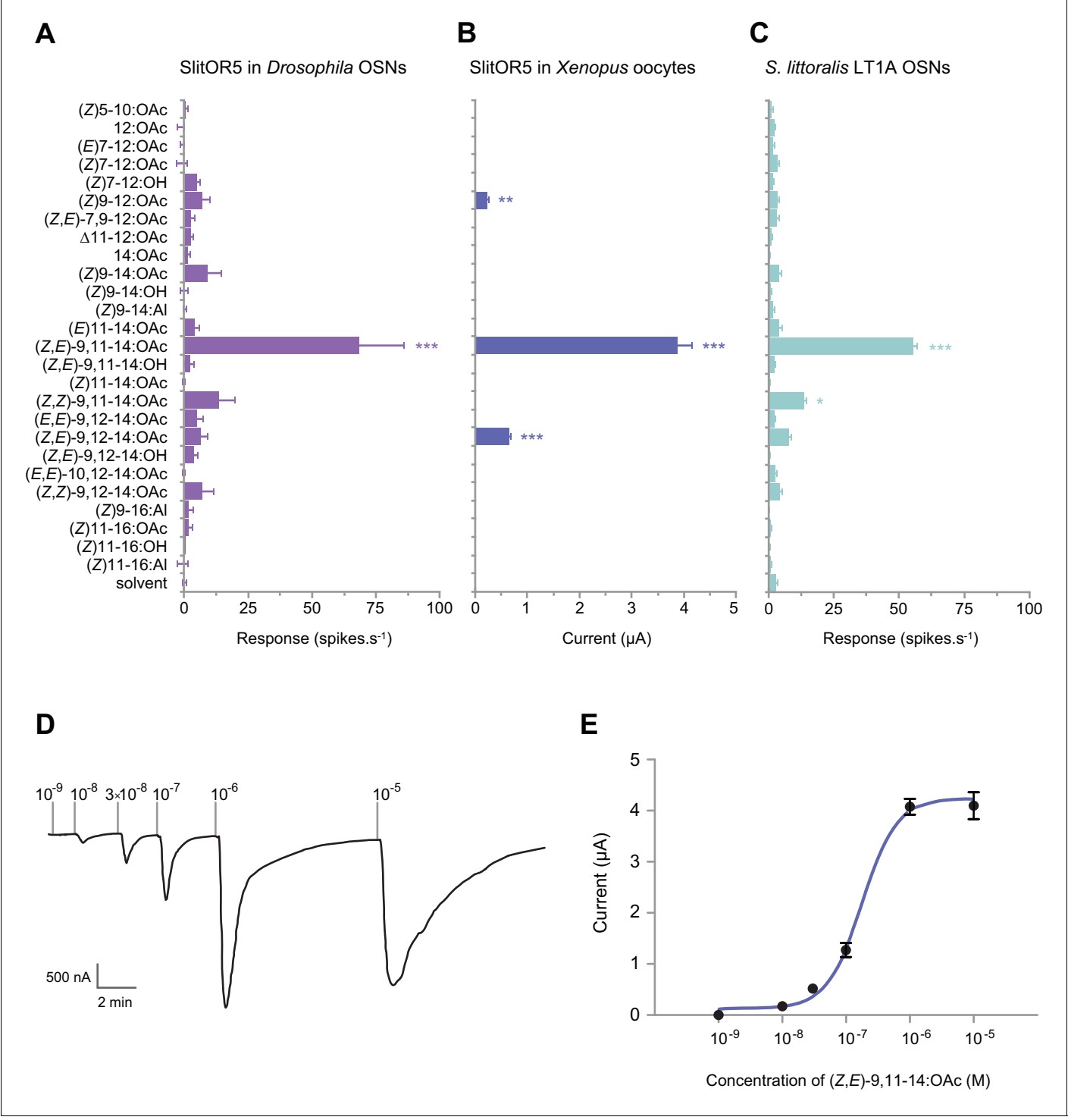

**Figure 2.** SlitOR5 is the receptor for the major component of the *S. littoralis* pheromone blend. (**A**) Action potential frequency of *Drosophila* at1 OSNs expressing SlitOR5 (*n* = 8) after stimulation with 26 type I pheromone compounds (10 µg loaded in the stimulus cartridge). ***p<0.001, significantly different from the response to solvent (one-way ANOVA followed by a Tukey's post hoc test). (**B**) Inward current measured in *Xenopus* oocytes co-expressing SlitOR5 and SlitOrco (*n* = 13–16) after stimulation with the same panel of pheromone compounds ($10^{-4}$ M solution). ***p<0.001, **p<0.01, significantly different from 0 (Wilcoxon signed rank test). (**C**) Action potential frequency of LT1A OSNs from *S. littoralis* male antennae (*n* = 8–16) after stimulation with pheromone compounds (1 µg loaded in the stimulus cartridge). ***p<0.001, *p<0.1, significantly different from the response to solvent (one-way ANOVA followed by a Tukey's post hoc test). (**D**) Representative trace showing the response of a *Xenopus* oocyte co-expressing SlitOR5 and SlitOrco after stimulation with a range of (*Z,E*)-9,11-14:OAc doses from $10^{-9}$ M to $10^{-5}$ M. (**E**) Dose-response curve of SlitOR5/Orco *Xenopus* oocytes

*Figure 2 continued on next page*

*Figure 2 continued*

($n$ = 9) stimulated with (Z,E)-9,11-14:OAc (EC$_{50}$ = $1.707 \times 10^{-7}$ M). Plotted values in (**A–C and E**) are mean responses ± SEM. Raw results for all experiments are available in *Figure 2—source data 1*.

The online version of this article includes the following source data for figure 2:

**Source data 1.** Raw results of electrophysiology experiments.

impaired in *SlitOr5$^{-/-}$* mutants, as these odorants elicited similar responses in wild-type, heterozygous and homozygous moths (*Figure 3B*).

Then, we analyzed the courtship behavior of *SlitOr5$^{-/-}$* and wild-type males in the presence of (Z, E)-9,11-14:OAc, and found a strong behavioral defect in mutants. Whereas more than 80% of wild-type males initiated a movement toward the pheromone source in the first 8 min, only a minority of *SlitOr5* mutants initiated such a movement (even after 30 min of test), a response similar to that of control wild-type males not stimulated with the pheromone (*Figure 3C*). All the steps of the courtship behavior were similarly affected (*Figure 3—figure supplement 1*). We also verified whether *SlitOr5* knock-out would result in mating inability. When paired with a wild-type virgin female, only 1 out of the 13 *SlitOr5$^{-/-}$* males tested was able to mate, compared to ~75% for wild-type males (*Figure 3D*). This behavioral defect was further confirmed by analyzing the number of eggs laid by the females and the number of offspring (*Figure 3E*). Overall, these results confirm that SlitOR5 is the receptor responsible for the detection of the major component of the *S. littoralis* female pheromone blend.

## A novel lineage of candidate moth pheromone receptors

In view of these results and the unexpected phylogenetic position of SlitOR5, we rebuilt the phylogeny of the lepidopteran OR clade containing SlitOR5 and the known receptors for type 0, type I and type II pheromones (highlighted in grey in *Figure 1B*), adding all ORs showing a strong sex-biased expression (at least 10-fold in one sex compared to the other) and ORs whose ligands were known as of September 2018 (*Supplementary file 2*). ORs grouped within eight different paralogous lineages, four of which including PRs (*Figure 4*). One was the so-called PR clade that, as previously observed, contained all type I PRs characterized so far (except SlitOR5) as well as two type II PRs (*Zhang et al., 2016*). The other three lineages harboring PRs consisted of one containing SlitOR5, one containing EgriOR31 (a type II PR from the geometrid *Ectropis grisescens*; *Li et al., 2017*) and one containing EsemOR3 and 5 (type 0 PRs from the non-dytrisian moth *Eriocrania semipurpurella*; *Yuvaraj et al., 2017*). Interestingly, most sex-biased lepidopteran ORs identified to date clustered within the PR clade and the lineage containing SlitOR5. While sex-biased ORs within the PR clade were mainly male-biased, the SlitOR5 lineage contained an equal proportion of male and female-biased receptors, identified from species belonging to different families of Lepidoptera. Deep nodes within the phylogeny were highly supported by the bootstrap analysis, enabling us to state that these two PR-containing clades were more closely related to clades harboring receptors for plant volatiles than to each other. This suggests that receptors tuned to type I pheromone compounds emerged twice independently during the evolution of Lepidoptera, and that the clade containing SlitOR5 may constitute a novel lineage of candidate PRs.

## Discussion

While moth sex pheromone receptors have been the most investigated ORs in Lepidoptera, with more than 60 being functionally characterized (*Zhang and Löfstedt, 2015*), it remains unclear how and when these specialized receptors arose. Type I PRs have been proposed to form a monophyletic, specialized clade of ORs, the so-called 'PR clade', which emerged early in the evolution of Lepidoptera (*Yuvaraj et al., 2017*; *Yuvaraj et al., 2018*). Here, we bring functional and phylogenetic evidence that type I PRs are not restricted to this clade and likely appeared twice independently in Lepidoptera. We focused on an atypical OR, SlitOR5, which exhibited a strong male-biased expression in antennae of the noctuid moth *S. littoralis* but did not group with the PR clade. We demonstrated, using a combination of heterologous expression and loss-of-function studies, that this OR is responsible for the detection of (Z,E)-9,11-14:OAc, the major component of the *S. littoralis* sex pheromone blend. Due to the unexpected phylogenetic position of SlitOR5 outside of the previously

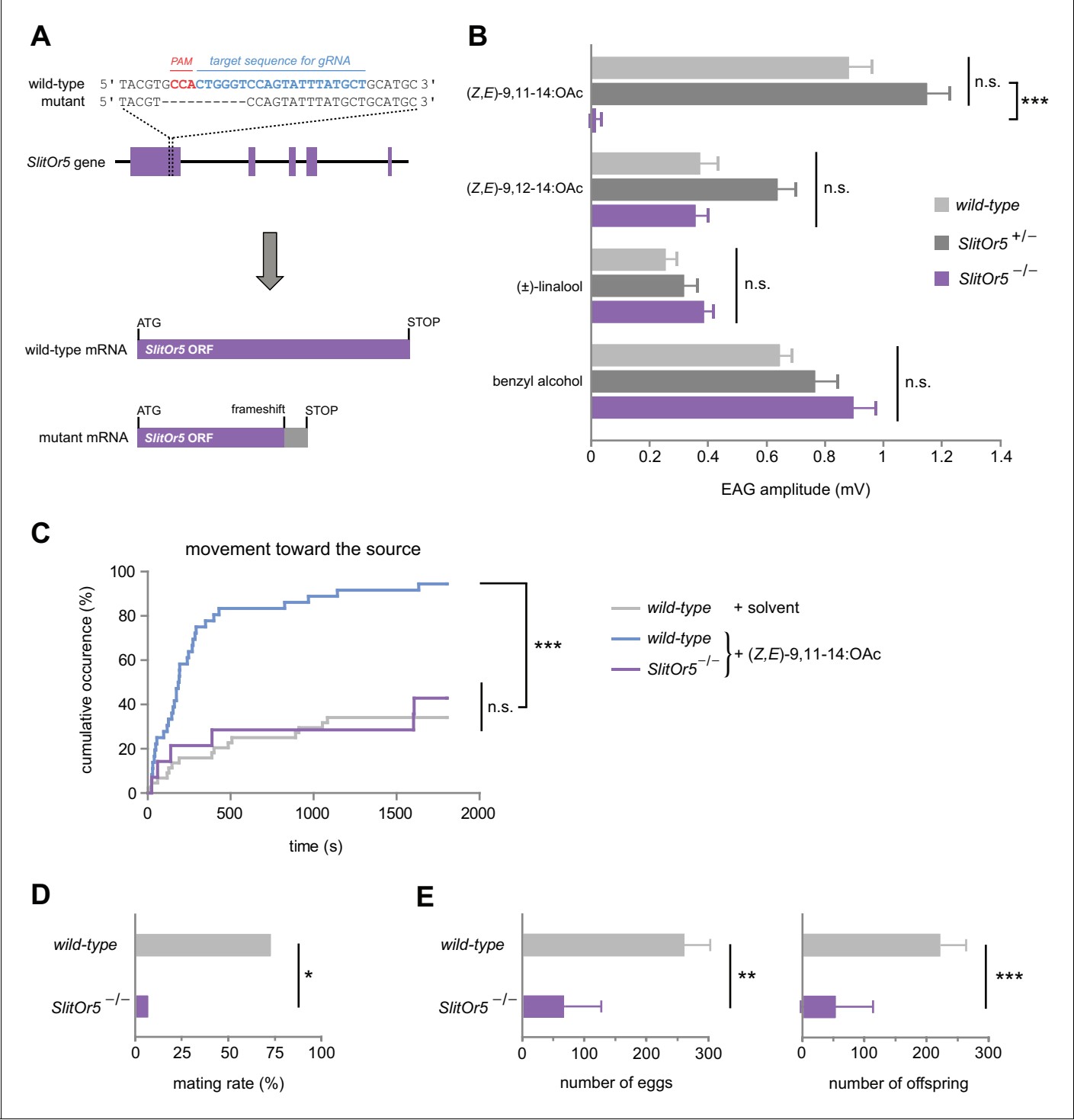

**Figure 3.** Response to the major pheromone component is abolished in *SlitOr5* mutants. (A) Location of the 10 bp deletion induced in the first exon of the *SlitOr5* gene by the CRISPR/Cas9 system. The sequence complementary to the RNA guide is indicated in blue, and the protospacer adjacent motif (PAM) in red. The frameshift created in the *SlitOr5* open-reading frame (ORF) induced a premature stop codon. (B) Electroantennogram (EAG) amplitude measured in *S. littoralis* male antennae isolated from wild-type animals (light grey, *n* = 14), heterozygous *SlitOr5* mutants (dark grey, *n* = 18) and homozygous *SlitOr5* mutants (purple, *n* = 8) after stimulation with pheromone compounds (1 μg in the stimulus cartridge) and plant volatiles (10 μg in the stimulus cartridge). Plotted values represent the normalized mean response ± SEM (response to the solvent was subtracted). ***p<0.001, significantly different from the response of the other genotypes; n.s.: not significantly different (one-way ANOVA, followed by a Tukey's post hoc test). Raw results for the EAG experiment are available in *Figure 3—source data 1*. (C) Cumulative proportion of *S. littoralis* males initiating a movement toward the odor source in homozygous *SlitOr5* mutants (purple, *n* = 14) stimulated with the major pheromone component (100 ng in the stimulus
*Figure 3 continued on next page*

*Figure 3 continued*

cartridge) and in wild-type animals stimulated with the pheromone (blue, *n* = 36) or with solvent alone (light grey, *n* = 44). ***p<0.001, significantly different from the other distributions; n.s.: not significantly different (log-rank test). Results obtained for other behavioral items are presented in *Figure 3—figure supplement 1*. (D) Proportion of wild-type *S. littoralis* males (light grey, *n* = 49) and homozygous *SlitOr5* mutants (purple, *n* = 13) that mated with a wild-type female during a period of 6 hr in the scotophase. *p<0.05, significant difference between the two genotypes (Fisher's exact test). (E) Number of eggs laid (left panel) and of offspring (right panel) obtained per female after the mating experiment. Plotted values represent the mean ± SEM. ***p<0.001, **p<0.005, significant difference between the two genotypes (Mann-Whitney *U* test). Raw results for all the behavioral experiments are available in *Figure 3—source data 2*.

The online version of this article includes the following source data and figure supplement(s) for figure 3:

**Source data 1.** Raw results of the EAG experiment.
**Source data 2.** Raw results of behavioral experiments.
**Figure supplement 1.** Behavioral response of wild-type and homozygous *SlitOr5* mutants to the major pheromone component.

---

defined PR clade, the question arose whether SlitOR5 is an exception or belongs to a previously unknown clade of moth PRs. This latter hypothesis is supported by the observation that the paralogous lineage containing SlitOR5 harbored many other sex-biased ORs, identified in species from six distinct lepidopteran families. Notably, male-biased ORs have been found in Lasiocampidae, Sphingidae, Noctuidae and Tortricidae. Among these, two ORs from *Ctenopseustis obliquana* and *C. herana* (Tortricidae) have been functionally studied by heterologous expression in cell cultures, but no ligand could be identified (*Steinwender et al., 2015*). In the Lasiocampidae species *Dendrolimus punctatus*, these male-biased ORs have been referred to as 'Dendrolimus-specific odorant receptors', with the suspicion that they would represent good PR candidates since in *Dendrolimus* species, there is no OR clustering in the PR clade (*Zhang et al., 2014*; *Zhang et al., 2017*). No functional data yet confirmed this suspicion.

Conversely, within the SlitOR5 lineage, almost half of the sex-biased ORs were female-biased (13 out of 28, compared to 4 out of 63 in the classical PR clade). Female-biased ORs have been generally proposed to be involved in the detection of plant-emitted oviposition cues, as demonstrated in *Bombyx mori* (*Anderson et al., 2009*). However, another interesting hypothesis is that they could be tuned to male sex pheromones. In moths, little attention has been put on male pheromones, which are known to be involved in various mating behaviors such as female attraction, female acceptance, aggregation of males to form leks, mate assessment or inhibition of other males (reviewed in *Conner and Iyengar, 2016*). The use of male pheromone systems has been selected multiple times in distinct moth families, as reflected by the chemical diversity of male pheromone compounds and of the disseminating structures (*Birch et al., 1990*; *Phelan, 1997*; *Conner and Iyengar, 2016*). It is thus expected that this polyphyletic nature of male pheromones would result in a large diversity of female PR types. Accordingly, female-biased ORs were found in different clades within the phylogeny. However, most remain orphan ORs, including BmorOR30 that does belong to the SlitOR5 lineage but for which no ligand could be identified (*Anderson et al., 2009*). Although the most common male-emitted volatiles are plant-derived pheromones (*Conner and Iyengar, 2016*), some male courtship pheromones are long-chained hydrocarbons related to type I female pheromone compounds (*Hillier and Vickers, 2004*) that could be detected by female-biased type I PRs such as those identified within the SlitOR5 lineage.

The ancestral protein from which the so-called 'PR clade' would have arisen is thought to be an OR tuned to plant-emitted volatiles (*Yuvaraj et al., 2017*; *Yuvaraj et al., 2018*). Here, we evidence that SlitOR5 is a new type I PR that belongs to a distinct early diverging lineage for which a role in pheromone detection had never been demonstrated. Together with the findings that PRs for Type 0 (*Yuvaraj et al., 2017*) and one PR for type II pheromones (*Li et al., 2017*) group in distinct paralogous lineages also unrelated to the PR clade, our data suggest that lepidopteran PRs have evolved four times in four paralogous lineages. Whether the SlitOR5 lineage has evolved from ORs that detected structurally related plant volatiles - as has been proposed for type 0 (*Yuvaraj et al., 2017*) and classical type I PRs - remains elusive. Yet, no OR tuned to plant volatiles has been identified in closely related lineages. More functional data on SlitOR5 paralogs and orthologs in different moth species, possibly revealing more type I PRs, will help in understanding evolutionary history of this lineage, as to when and how these receptors have evolved, and confirm that we are facing a novel type I PR clade.

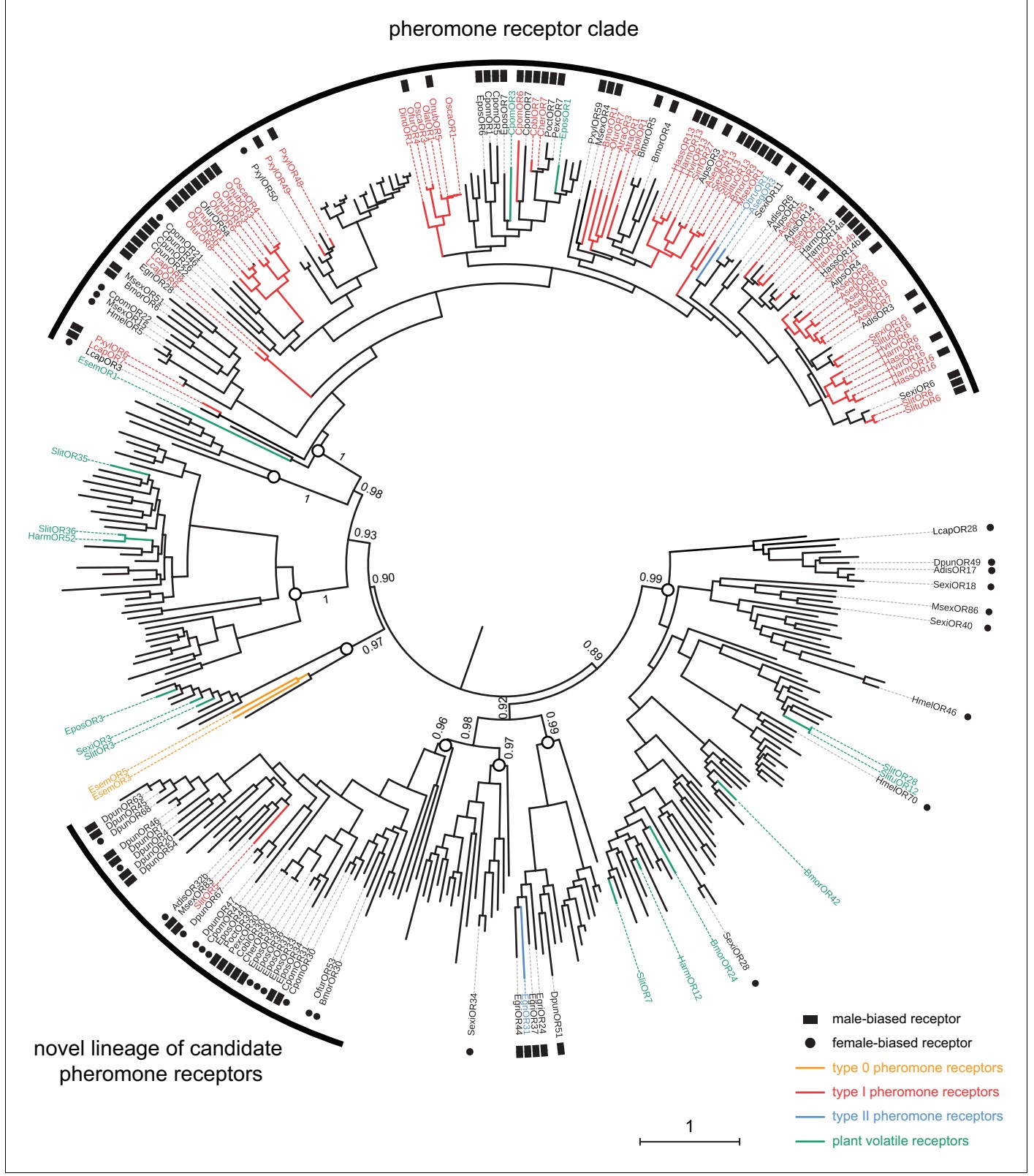

**Figure 4.** SlitOR5 may define a novel lineage of candidate pheromone receptors in Lepidoptera. Maximum likelihood phylogeny of the lepidopteran OR clade that includes all the paralogous lineages containing pheromone receptors. 360 sequences from 34 lepidopteran species were included. Functional and expression data shown on the figure have been compiled from the literature (*Supplementary file 2*). Branch colors indicate OR function, when characterized: PRs for type I pheromones are depicted in red, those for type II pheromones in blue and those for type 0 pheromones in

*Figure 4 continued*

orange. ORs tuned to plant volatiles are depicted in green. Symbols at the edge indicate expression data: male-biased ORs are highlighted with black squares and female-biased ORs with black dots. Circles on the nodes indicate the distinct paralogous OR lineages. Support values on basal nodes are transfer bootstrap expectation (TBE) values. The tree has been rooted using an outgroup as identified in the lepidopteran OR phylogeny shown in *Figure 1*. The scale bar indicates the expected number of amino acid substitutions per site. The sequence alignment file is available in *Figure 4— source data 1*.

The online version of this article includes the following source data for figure 4:

**Source data 1.** Alignment of amino acid sequences used to build the phylogeny (FASTA format).

# Materials and methods

## Key resources table

| Reagent type (species) or resource | Designation | Source or reference | Identifiers | Additional information |
|---|---|---|---|---|
| Gene (*Spodoptera littoralis*) | *SlitOr5* | GenBank | GB:MK614705 | |
| Gene (*Spodoptera littoralis*) | *SlitOrco* | *Malpel et al. (2008)*; PMID:18828844; GenBank | GB:EF395366 | |
| Genetic reagent (*Drosophila melanogaster*) | Or67d<sup>GAL4</sup> | *Kurtovic et al. (2007)*; PMID: 17392786 | FLYB:FBal0210948 | kindly provided by B. Dickson |
| Genetic reagent (*Drosophila melanogaster*) | y<sup>1</sup> M{vas-int.Dm}ZH-2A w*; M{3xP3-RFP.attP}ZH-51C | *Bischof et al. (2007)*; PMID: 17360644; Bloomington Drosophila Stock Center | BDSC:24482 | |
| Genetic reagent (*Drosophila melanogaster*) | UAS-*SlitOr5* | This study | | See Materials and methods |
| Recombinant DNA reagent | pUAST.attB (plasmid) | *Bischof et al. (2007)*; PMID: 17360644; GenBank | GB:EF362409 | kindly provided by J. Bischof |
| Recombinant DNA reagent | pUAST.attB-*SlitOr5* (plasmid) | This study | | See Materials and methods |
| Recombinant DNA reagent | pCS2+ (plasmid) | *Turner and Weintraub (1994)*; PMID: 7926743 | | kindly provided by C. Héligon |
| Recombinant DNA reagent | pCS2+-*SlitOr5* (plasmid) | This study | | See Materials and methods |
| Recombinant DNA reagent | pCS2+-*SlitOrco* (plasmid) | This study | | See Materials and methods |
| Sequence-based reagent | Or5up | This study | PCR primers | TCGGGAGAAACTGAAGGACGTTGT |
| Sequence-based reagent | Or5do | This study | PCR primers | GCACGGAACCGCACTTATCACTAT |
| Sequence-based reagent | Rpl13up | This study | PCR primers | GTACCTGCCGCTCTCCGTGT |
| Sequence-based reagent | Rpl13do | This study | PCR primers | CTGCGGTGAATGGTGCTGTC |
| Sequence-based reagent | *SlitOr5* guide RNA | This study | gRNA | AGCATAAATACTGGACCCAGTGG |
| Sequence-based reagent | Or5_forward | This study | PCR primers | CCAAAAGGACTTGGACTTTGAA |
| Sequence-based reagent | Or5_reverse | This study | PCR primers | CCCGAATCTTTTCAGGATTAGAA |

## Animal rearing and chemicals

*S. littoralis* were reared in the laboratory on a semi-artificial diet (*Poitout and Buès, 1974*) at 22℃, 60% relative humidity and under a 16 hr light:8 hr dark cycle. Males and females were sexed as pupae and further reared separately. *D. melanogaster* lines were reared on standard cornmeal-yeast-agar medium and kept in a climate- and light-controlled environment (25℃, 12 hr light:12 hr dark cycle). The 26 pheromone compounds used for electrophysiology experiments (*Supplementary file 1*) were either synthesized in the lab or purchased from Sigma-Aldrich (St Louis, MO) and Pherobank (Wijk bij Duurstede, The Netherlands). Paraffin oil was purchased from Sigma-Aldrich and hexane from Carlo Erba Reagents (Val de Reuil, France).

## Quantitative real-time PCR

Total RNA from three biological replicates of 15 pairs of antennae of two-day-old virgin male and female *S. littoralis* was extracted using RNeasy Micro Kit (Qiagen, Hilden, Germany), which included a DNase treatment. cDNA was synthesized from total RNA (1 µg) using Invitrogen SuperScript II reverse transcriptase (Thermo Fisher Scientific, Waltham, MA). Gene-specific primers were designed for *SlitOr5* (*Or5up*: 5'-TCGGGAGAAACTGAAGGACGTTGT-3', *Or5do*: 5'-GCACGGAACCGCAC TTATCACTAT-3') and for the reference gene *SlitRpl13* (*Rpl13up*: 5'-GTACCTGCCGCTCTCCGTGT-3', *Rpl13do*: 5'-CTGCGGTGAATGGTGCTGTC-3'). qPCR mix was prepared in a total volume of 10 µL with 5 µL of LightCycler 480 SYBR Green I Master (Roche, Basel, Switzerland), 4 µL of diluted cDNA (or water for the negative control) and 0.5 µM of each primer. qPCR assays were performed using the LightCycler 480 Real-Time PCR system (Roche). All reactions were performed in triplicate for the three biological replicates. The PCR program began with a cycle at 95℃ for 13.5 min, followed by 50 cycles of 10 s at 95℃, 15 s at 60℃ and 15 s at 72℃. Dissociation curves of the amplified products were performed by gradual heating from 55℃ to 95℃ at 0.5 ℃.s$^{-1}$. A negative control (mix without cDNA) and a fivefold dilution series protocol of pooled cDNAs (from all conditions) were included. The fivefold dilution series were used to construct relative standard curves to determine the PCR efficiencies used for further quantification analyses. Data were analyzed with the Light-Cycler 480 software (Roche). Normalized expression of the *SlitOr5* gene was calculated with the Q-Gene software (*Joehanes and Nelson, 2008*) using *SlitRpl13* as a reference, considering it displays consistent expression as previously described in *Durand et al. (2010)*.

## Heterologous expression of SlitOR5 in *Drosophila*

The *SlitOr5* full-length open-reading frame (1191 bp, GenBank acc. num. MK614705) was subcloned into the pUAST.attB vector. Transformant *UAS-SlitOr5* balanced fly lines were generated by Best-Gene Inc (Chino Hills, CA), by injecting the pUAST.attB-*SlitOr5* plasmid (Endofree prepared, Qiagen) into fly embryos with the genotype $y^1$ *M{vas-int.Dm}ZH-2A* *w\**; *M{3xP3-RFP.attP}ZH-51C* (*Bischof et al., 2007*), leading to a non-random insertion of the *UAS-SlitOr5* construct into the locus 51C of the second chromosome. The *UAS-SlitOr5* balanced line was then crossed to the *Or67d*$^{GAL4}$ line (*Kurtovic et al., 2007*) to obtain double homozygous flies (genotype *w; UAS-SlitOr5,w$^+$; Or67d-*$^{GAL4}$) expressing the *SlitOr5* transgene in at1 OSNs instead of the endogenous *Drosophila* receptor gene *Or67d*. The correct expression of *SlitOr5* was confirmed by RT-PCR on total RNA extracted from 100 pairs of antennae.

## Single-sensillum recordings

Single-sensillum extracellular recordings on *Drosophila* at1 OSNs were performed as previously described (*de Fouchier et al., 2015*). OSNs were stimulated during 500 ms, using stimulus cartridges containing 10 µg of pheromone (1 µg/µl in hexane) dropped onto a filter paper. Single-sensillum recordings on *S. littoralis* LT1A OSNs were performed using the tip-recording technique, as previously described (*Pézier et al., 2007*). Briefly, the tips of a few LT1 sensilla were cut off using sharpened forceps and a recording glass electrode filled with a saline solution (170 mM KCl, 25 mM NaCl, 3 mM MgCl$_2$, 6 mM CaCl$_2$, 10 mM HEPES and 7.5 mM glucose, pH 6.5) was slipped over the end of a cut LT1 sensillum. OSNs were stimulated with an air pulse of 200 ms (10 L.h$^{-1}$), odorized using a stimulus cartridge containing 1 µg of pheromone (diluted at 1 µg/µl in hexane). Odorants were considered as active if the response they elicited was statistically different from the response elicited by the solvent alone (one-way ANOVA followed by a Tukey's *post hoc* test).

## Heterologous expression of SlitOR5 in *Xenopus* oocytes and two-electrode voltage-clamp recordings

Open-reading frames of *SlitOr5* and *SlitOrco* (GenBank acc. num. EF395366, *Malpel et al., 2008*) were subcloned into the pCS2+ vector (*Turner and Weintraub, 1994*). Template plasmids were fully linearized with PteI for pCS2+-*SlitOr5* and NotI for pCS2+-*SlitOrco* and capped cRNAs were transcribed using SP6 RNA polymerase. Purified cRNAs were re-suspended in nuclease-free water at a concentration of 2 µg/µL and stored at −80°C. Mature healthy oocytes were treated with 2 mg/ml collagenase type I in washing buffer (96 mM NaCl, 2 mM KCl, 5 mM MgCl$_2$ and 5 mM HEPES, pH 7.6) for 1–2 hr at room temperature. Oocytes were later microinjected with 27.6 ng of *SlitOr5* cRNA and 27.6 ng of *SlitOrco* cRNA. After 4 days of incubation at 18°C in 1 × Ringer's solution (96 mM NaCl, 2 mM KCl, 5 mM MgCl2, 0.8 mM CaCl2, and 5 mM HEPES, pH 7.6) supplemented with 5% dialyzed horse serum, 50 mg/ml tetracycline, 100 mg/ml streptomycin and 550 mg/ml sodium pyruvate, the whole-cell currents were recorded from the injected oocytes with a two-electrode voltage clamp. Oocytes were exposed to the 26 pheromone compounds diluted at $10^{-4}$ M in Ringer's solution, with an interval between exposures which allowed the current to return to baseline. Data acquisition and analysis were carried out with Digidata 1440A and pCLAMP10 software (Molecular Devices, San Jose, CA). Odorants were considered as active if the mean response they elicited was statistically different from 0 (Wilcoxon signed rank test). Dose–response experiments were performed using pheromone concentrations ranging from $10^{-9}$ up to $10^{-5}$ M and data were analyzed using GraphPad Prism 5.

## *SlitOr5* knock-out via CRISPR/Cas9

A guide RNA (gRNA sequence: AGCATAAATACTGGACCCAG TGG) was designed against the first exon of the *SlitOr5* gene using the CRISPOR gRNA design tool (crispor.tefor.net; *Haeussler et al., 2016*) and transcribed after subcloning into the DR274 vector using the HiScribe T7 High Yield RNA Synthesis Kit (New England Biolabs, Ipswich, MA). The Cas9 protein was produced in *Escherichia coli* as previously described (*Ménoret et al., 2015*). A mix of Cas9 protein and gRNA was injected in freshly laid eggs using an Eppendorf - Transjector 5246, as previously described (*Koutroumpa et al., 2016*). Individual genotyping at every generation was performed via PCR on genomic DNA extracted from larvae pseudopods (Wizard Genomic DNA Purification Kit, Promega, Madison, WI) using gene-specific primers (*Or5*_forward: 5'-CCAAAAGGACTTGGACTTTGAA-3'; *Or5*_reverse: 5'-CCCGAATCTTTTCAGGATTAGAA-3') amplifying a fragment of 728 bp encompassing the target sequence. Mutagenic events were detected by sequencing the amplification products (Biofidal, Vaulx-en-Velin, France). G0 larvae carrying a single mutagenic event were reared until adults and crossed with wild-type individuals. Homozygous G2 individuals were obtained by crossing G1 heterozygous males and females.

## Phenotyping of CRISPR/Cas9 mutants by electroantennogram recordings

Electroantennogram recordings were performed as previously described (*Koutroumpa et al., 2016*) on isolated male antennae from wild-type animals and from heterozygous and homozygous *SlitOr5* mutants. Antennae were stimulated using stimulus cartridges loaded with 10 µg of linalool or benzyl alcohol diluted in paraffin oil, and 1 µg of (*Z,E*)-9,11-14:OAc or (*Z,E*)-9,12-14:OAc diluted in hexane. Stimulations lasted for 500 ms (30 L/h). Negative controls consisted of paraffin oil and hexane alone. The maximum depolarization amplitude was measured using the pCLAMP10 software. Normalized mean responses were calculated (response to the solvent was subtracted) and data were analyzed using a one-way ANOVA followed by a Tukey's post hoc test.

## Behavioral experiments

For courtship monitoring, 2-day-old wild-type or homozygous *SlitOr5* mutant males were placed individually in a plastic squared petri dish (size = 12 cm) before the onset of the scotophase, and experiments started at the middle of the scotophase (22°C, 70% relative humidity). Odorant stimulation was performed using Pasteur pipettes containing a filter paper loaded with 100 ng of (*Z,E*)-9,11-14:OAc (10 ng/µl in hexane), or hexane alone as control. The narrow end of the pipette was inserted into the petri dish, and a constant stream of charcoal-filtered, humidified air (0.2 L.min$^{-1}$) passed

through the pipette during all the experiment. Male courtship behavior was recorded under dim red light during 30 min using a webcam (Logitech QuickCam Pro 9000). The latency of each of the following stereotyped behavioral items was individually screened: antennal flicking, movement toward the odor source, wing fanning, abdomen curving and extrusion of genitalia. Results were presented as cumulative occurrence of each item along the time of the experiment. Statistical analysis of the distributions obtained for each treatment were compared using a log-rank (Mantel-Cox) test. For mating experiments, 2-day-old wild-type or homozygous *SlitOr5* mutant males were paired with a 2-day-old wild-type virgin female in a cylindrical plastic box (diameter 8 cm, height 5 cm), the walls of which were covered up with filter paper. Experiments started at the onset of the scotophase and lasted over 6 hr. Copulation events were visually inspected every 20 min under dim red light. After the end of the experiment, females were kept in the boxes for 12 hr. Then, females were discarded and filter papers were examined for the presence of egg clutches. When present, eggs were transferred to a rearing plastic box, hatch was monitored every day during 7 days and second-instar larvae were counted, if any. Statistical analysis of mating rates was done using a Fisher's exact test, and analyses of the number of eggs and offspring were done using a Mann-Whitney *U* test.

## Phylogenetic analyses

The dataset of lepidopteran amino acid OR sequences used to build the phylogeny shown in *Figure 1* included entire OR repertoires from the following nine species, each belonging to a different lepidopteran superfamily: *Bombyx mori* (Bombycoidea; *Tanaka et al., 2009*), *Dendrolimus punctatus* (Lasiocampoidea; *Zhang et al., 2017*), *Ectropis grisescens* (Geometroidea; *Li et al., 2017*), *Epiphyas postvittana* (Tortricoidea; *Corcoran et al., 2015*), *Eriocrania semipurpurella* (Eriocranioidea; *Yuvaraj et al., 2017*), *Heliconius melpomene* (Papilionoidea; *Heliconius Genome Consortium, 2012*), *Ostrinia furnacalis* (Pyraloidea; *Yang et al., 2015*), *Plutella xylostella* (Yponomeutoidea; *Engsontia et al., 2014*) and *Spodoptera littoralis* (Noctuoidea; *Walker et al., 2019*). The dataset used to build the phylogeny shown in *Figure 4* contained amino acid sequences from the same nine species falling into that clade, plus all the sequences of ORs within that clade showing a marked sex-biased expression (threshold of a 10-fold difference in expression rate between male and female antennae, based on RNAseq or qPCR experiments) and/or for which ligands have been identified as of September 2018 (*Supplementary file 2*). Alignments were performed using Muscle (*Edgar, 2004*) as implemented in Jalview v2.10.5 (*Waterhouse et al., 2009*). Phylogenetic reconstruction was performed using maximum likelihood. The best-fit model of protein evolution was selected by SMS (*Lefort et al., 2017*) and tree reconstruction was performed using PhyML 3.0 (*Guindon et al., 2010*). Node support was first assessed using 100 bootstrap iterations, then the file containing bootstrap trees was uploaded on the Booster website (*Lemoine et al., 2018*) to obtain the TBE (Transfer Bootstrap Expectation) node support estimation. Figures were created using the iTOL web server (*Letunic and Bork, 2016*).

## Acknowledgements

We are grateful to Lixiao Du and Françoise Bozzolan for technical assistance, Bin Yang for his help and advice, and Pascal Roskam and Philippe Touton for insect rearing. We also thank Christophe Héligon (CRB Xénope, Rennes) for providing the pCS2+ plasmid. This work has been funded by the French National Research Agency (ANR-16-CE02-0003-01 and ANR-16-CE21-0002-01 grants), the National Natural Science Foundation of China (31725023, 31621064) and a PRC NSFC-CNRS 2019 grant. Research was conducted as part of the CAAS-INRA Associated International Laboratory in Plant Protection.

## Additional information

### Competing interests

Arthur de Fouchier, Emmanuelle Jacquin-Joly, Nicolas Montagné: The intellectual property rights of SlitOR5 have been licensed by Inra, Sorbonne Université and CNRS for the purpose of developing novel insect control agents. The other authors declare that no competing interests exist.

## Funding

| Funder | Grant reference number | Author |
|---|---|---|
| Agence Nationale de la Recherche | ANR-16-CE02-0003-01 | Nicolas Montagné |
| Agence Nationale de la Recherche | ANR-16-CE21-0002-01 | Emmanuelle Jacquin-Joly |
| National Natural Science Foundation of China | 31725023 | Guirong Wang |
| National Natural Science Foundation of China | 31621064 | Guirong Wang |
| PRC NSFC-CNRS | | Guirong Wang |

The funders had no role in study design, data collection and interpretation, or the decision to submit the work for publication.

## Author contributions

Lucie Bastin-Héline, Data curation, Investigation, Writing—original draft; Arthur de Fouchier, William B Walker III, Investigation, Writing—review and editing; Song Cao, Data curation, Investigation, Writing—review and editing; Fotini Koutroumpa, Gabriela Caballero-Vidal, Stefania Robakiewicz, Marie-Christine François, Tatiana Ribeyre, Annick Maria, Thomas Chertemps, Investigation; Christelle Monsempes, Data curation, Investigation; Anne de Cian, Resources; Guirong Wang, Conceptualization, Funding acquisition, Writing—review and editing; Emmanuelle Jacquin-Joly, Conceptualization, Funding acquisition, Writing—original draft; Nicolas Montagné, Conceptualization, Data curation, Funding acquisition, Writing—original draft

## Author ORCIDs

William B Walker III  https://orcid.org/0000-0003-2798-9616
Emmanuelle Jacquin-Joly  http://orcid.org/0000-0002-6904-2036
Nicolas Montagné  https://orcid.org/0000-0001-8810-3853

## Decision letter and Author response

Decision letter https://doi.org/10.7554/eLife.49826.sa1
Author response https://doi.org/10.7554/eLife.49826.sa2

## Additional files

### Supplementary files

• Supplementary file 1. List of synthetic compounds used for electrophysiology experiments.

• Supplementary file 2. Functional and sex-biased expression data available for lepidopteran pheromone receptors (as of September 2018).

• Transparent reporting form

### Data availability

All data generated or analysed during this study are included in the manuscript and supporting files. Source data files have been provided for Figures 1, 2, 3 and 4.

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
