## [Decision Letter]

**Acceptance summary:**

Male moths travel extraordinary distances to seek a female mate. Pheromone blends emitted by females are essential for all aspects of male mating behavior. In this study, Bastin-Heline et al. characterized an odorant receptor (SlitOR5) in *S. littoralis* which is highly expressed in male antennae and found that it specifically responds to the major sex pheromone component. Male moths with CRISPR-generated mutations in SlitOR5 fail to detect the major pheromone by electroantennogram recordings and have severe defects in pheromone-mediated attraction and in all aspects of courtship behavior. Phylogenetic analysis demonstrated that SlitOR5 is on a separate clade from the classic Pheromone Receptor clades, suggesting it marks a new clade in lepidoptera. This study is significant because it provides a dramatic demonstration of how a single pheromone receptor impacts animal behavior. This is an important advance for the field of chemical ecology of insects and will attract a general audience interested in sensory-guided behavior.

**Decision letter after peer review:**

Thank you for submitting your article "A novel lineage of candidate pheromone receptors for sex communication in moths" for consideration by *eLife*. Your article has been reviewed by three peer reviewers, and the evaluation has been overseen by Kristin Scott as the Reviewing Editor and K VijayRaghavan as the Senior Editor. The following individuals involved in review of your submission have agreed to reveal their identity: Marcus C Stensmyr (Reviewer #1); Laurence J Zwiebel (Reviewer #2); Wei Xu (Reviewer #3).

The reviewers have discussed the reviews with one another and the Reviewing Editor has drafted this decision to help you prepare a revised submission.

Summary:

Bastin-Heline et al. identified a new odorant receptor (SlitOR5) in *S. littoralis* which is highly expressed in male antennae and specifically responds to the major sex pheromone component. The functional characterization and knockout studies are solid, well-controlled studies that provide nice data in support of SlitOR5 being selectively tuned to the (*Z,E*)-9,11-14:OAc pheromone and responsible for the electrophysiological sensitivity. The paper is well written and easy to follow. The experiments (except the behavioral ones) are all done to a high technical standard and support the conclusions of the manuscript. The findings presented will certainly be met with interest in the moth/pheromone community and opens up new research venues.

Essential revisions:

Given that the authors have gone through the extensive trouble of generating a mutant line, I'm surprised not more effort was spent in analyzing these animals behaviorally. The authors should, at bare minimum, quantify the behavioral deficiency they report and present this in a figure or table. Preferably, they should add additional behavioral tests, e.g. wind tunnel (assuming these animals are able to fly that is).

---

## [Author Response]

Essential revisions:Given that the authors have gone through the extensive trouble of generating a mutant line, I'm surprised not more effort was spent in analyzing these animals behaviorally. The authors should, at bare minimum, quantify the behavioral deficiency they report and present this in a figure or table. Preferably, they should add additional behavioral tests, e.g. wind tunnel (assuming these animals are able to fly that is).

We agree with the reviewers. In the previous version of the manuscript, we were unfortunately not able to provide results of a more thorough behavioral analysis. The main reason is that we were not able to get enough homozygous mutant males, due to the impossibility to establish an OR5 homozygous mutant strain (males do not mate). During the last months, we could generate more mutants and we carried out two behavioral experiments: 1) we made novel mating tests (n=13) between a mutant OR5 male and a wild-type female and analyzed their ability to mate, the number of laid eggs and the number of offspring, compared with wild-type male/female couples. 2) we quantified several behavioral items (using video tracking) on mutant and wild-type males in response to stimulation by the major pheromone component Z9E11-14:Ac. We present the results of these experiments in this revised version of the manuscript (included in a novel version of Figure 3 and Figure 3—figure supplement 1), and we now clearly show that the OR5 mutation causes strong behavioral defects and that activation of this receptor is mandatory for initiating courtship and mating. The text of the “Results” and “Materials and methods” sections has been modified accordingly. The supplementary behavioral experiments have been carried out by two colleagues who have been added to the list of authors in this revised version of the manuscript.